# Role of point-of-care tests in the management of febrile children: a qualitative study of hospital-based doctors and nurses in England

Edmond Li [1,2] Juan Emmanuel Dewez [1] Queena Luu,[1] Marieke Emonts,[3] Ian Maconochie,[4] Ruud Nijman,[5] Shunmay Yeung [1,6]

EL, JED and QL contributed equally.

EL, JED and QL are joint first authors.

For numbered affiliations see end of article.

**Correspondence to**
Professor Shunmay Yeung; shunmay.yeung@lshtm.ac.uk

## ABSTRACT

**Objectives** The use of rapid point-of-care tests (POCTs) has been advocated for improving patient management and outcomes and for optimising antibiotic prescribing. However, few studies have explored healthcare workers' views about their use in febrile children. The aim of this study was to explore the perceptions of hospital-based doctors and nurses regarding the use of POCTs in England.

**Study design** Qualitative in-depth interviews with purposively selected hospital doctors and nurses. Data were analysed thematically.

**Setting** Two university teaching hospitals in London and Newcastle.

**Participants** 24 participants (paediatricians, emergency department doctors, trainee paediatricians and nurses).

**Results** There were diverse views about the use of POCTs in febrile children. The reported advantages included their ease of use and the rapid availability of results. They were seen to contribute to faster clinical decision-making; the targeting of antibiotic use; improvements in patient care, flow and monitoring; cohorting (ie, the physical clustering of hospitalised patients with the same infection to limit spread) and enhancing communication with parents. These advantages were less evident when the turnaround for results of laboratory tests was 1–2 hours. Factors such as clinical experience and specialty, as well as the availability of guidelines recommending POCT use, were also perceived as influential. However, in addition to their perceived inaccuracy, participants were concerned about POCTs not resolving diagnostic uncertainty or altering clinical management, leading to a commonly expressed preference for relying on clinical skills rather than test results solely.

**Conclusion** In this study conducted at two university teaching hospitals in England, participants expressed mixed opinions about the utility of current POCTs in the management of febrile children. Understanding the current clinical decision-making process and the specific needs and preferences of clinicians in different settings will be critical in ensuring the optimal design and deployment of current and future tests.

## INTRODUCTION

Febrile illness is one of the most common reasons for children to present to hospital,

### Strengths and limitations of this study

► A specific focus on the paediatric population.
► The inclusion of two hospitals in two different settings.
► A multidisciplinary and diverse group of participants including nurses and doctors with different levels of clinical experience and training.
► The study took place in university teaching hospitals and did not explore the views of clinicians in other settings.
► The study explored the views of participants on several point-of-care tests (POCTs) used in febrile children, limiting the amount of detail obtained about specific POCTs.

and managing it can often be challenging.[1 2] The majority of febrile children will have self-limiting infections that can be safely managed at home, often without antibiotics.[3–5] However, a few will have potentially severe infections and identifying these children from the others can be difficult, as clinical symptoms and signs are often nonspecific in children.[3] Consequently, they may be prescribed antibiotics that are not needed, subjected to invasive tests, and admitted for monitoring while awaiting microbiology results.[4 5] As well as the direct impact this may have on children and parents in terms of distress, costs and inconvenience,[6] there may also be indirect impacts in terms of antimicrobial resistance.[7]

The WHO advocates the use of rapid point-of-care tests (POCTs) to reduce the use of antibiotics.[8] POCTs have the potential to aid clinical decision-making, limit the use of other invasive tests and improve the use of medical resources in general.[9] However, achieving these impacts depends on several factors, including whether POCTs are taken up by healthcare workers and how they influence clinical decisions.[10] This cannot be

BMJ

assumed, for example, when malaria POCTs were introduced, many clinicians continued to prescribe antimalarials despite patients testing negative.[11]

The types of POCTs that are currently available for the management of childhood infections include urine dipsticks; rapid tests for *Group A Streptococcal* throat infections (rapid strep test); rapid respiratory virus tests for respiratory syncytial virus (RSV) or influenza and blood-based tests including C reactive protein (CRP) and blood gas analysers.

To date, research exploring the views of clinicians on the use of POCTs for managing patients with infections, have predominantly focused on general practitioners (GPs) treating adult patients.[12–19] To the best of our knowledge, only one study pertains to the perceptions of hospital-based healthcare workers regarding the use of POCTs in children.[20]

The aim of this study was to explore the experiences and attitudes of hospital-based doctors and nurses regarding the use of POCTs in children with acute infections and to identify factors that influence their use.

## METHODS

Qualitative in-depth interviews with doctors and nurses were conducted to better understand their experiences and attitudes regarding POCT use in children with suspected infection.[21] Ontologically, our approach adhered to subtle realism, and epistemologically to interpretivism,[22 23] because we believed that although reality (ie, the use of diagnostics) exists independently of our beliefs, the interpretation of participants' perceptions and how they make sense of their experiences were needed to ascertain what influences POCT usage.

The research team comprised of two public health professionals and five paediatricians with experience in qualitative methods. Two topic guides, one for doctors and one for nurses, were developed based on a review of the literature (online supplemental file 1). These were piloted with two paediatricians and two nurses at St Mary's Hospital in London. The pilot interviews were not included in the analysis. The topic guides were iteratively adapted during data collection after regular debriefings between EL, QL and JED to discuss main findings, refine questions and explore new areas of inquiry.[24]

Eligible participants comprised of doctors and nurses with responsibility for making decisions about whether or not to use POCTs on febrile children in two university teaching hospitals: the Great North Children's Hospital in Newcastle and St Mary's Hospital in London. We purposively chose not to interview other subspecialty paediatricians such as paediatric infectious disease specialists as they are less often responsible for decisions on POCT use in the acute hospital setting. Purposive maximum variation sampling was used, focusing on participants with different clinical roles and varying years of experience. A sampling matrix was developed specifying the targeted number of participants of each healthcare worker category (consultant paediatrician, paediatric trainee with three or more years training, paediatric trainees with less than three years training and emergency department (ED) nurses) (online supplemental file 2). As members of staff of the two hospitals, SY, RN, IM, ME contacted available paediatric staff who fitted the inclusion criteria to ascertain their interest in being interviewed for the study. Those who agreed were then followed-up by the two interviewers (EL, male) and (QL, female) who were not known to the participants, and who provided the detailed participant information sheet obtained written informed consent and arranged the interview date. The study aimed to include 22–30 participants, a sample size that is in line with other qualitative studies investigating healthcare workers' perceptions and was expected to allow for data saturation.[12 13 18 25] Identifying the actual point of data saturation was important to make best use of the available time and resources and to minimise the disruption caused by diverting healthcare providers from their normal clinical activities. During the regular debriefings, EL, QL, and JED identified whether any new ideas had been raised. Once no new ideas had emerged from the last several interviews, the authors agreed that data saturation had been reached. One-on-one audio-recorded interviews took place between June and August 2018 during normal work hours at the respondents' workplace. Either EL or QL conducted the interview while the other took field notes.

The interview records were transcribed verbatim by EL and QL and back checked against the recording and field notes for accuracy. The transcripts were anonymised (only participants' job titles were documented) and then analysed thematically.[24] The analysis was both deductive (ie, partly based on pre-existing knowledge) and inductive (ie, additional concepts were identified from the data set). EL and QL generated two initial lists of coding from the data set through line-by-line coding (online supplemental file 3). EL, QL and JED then agreed on an analytical framework comprising the most relevant codes and used the framework to recode the entire data set. Matrices were used to display data related to initial themes (online supplemental file 4). Iterative exploration of the data identified patterns within the data set and continued until no new themes were identified. Further refinement was carried out with the rest of the research team to ensure themes were internally coherent and distinctive to produce clear descriptions and explanations of each theme.

Ethical approval was obtained from the London School of Hygiene & Tropical Medicine Ethics Committee (Reference: 15 040-15088). Health Research Authority approval (Reference: IRAS 248723) and administrative approval were obtained from the Imperial College Healthcare NHS Trusts (Reference: 18SM4662) and Newcastle upon Tyne Hospital NHS Foundation (Reference: 248723). The Consolidated Criteria for Reporting Qualitative Research (COREQ) checklist has been completed (online supplemental file 5).

**Table 1** Number of different types of healthcare workers interviewed

| Healthcare workers | Hospital site | | Total |
| | SMH, London | GNCH, Newcastle | |
|---|---|---|---|
| Nurses—Paediatric Emergency Department | 1 | 3 | 4 |
| Consultants—Paediatric Emergency Department | 3 | 2 | 5 |
| Consultants—General Paediatrics | 2 | 1 | 3 |
| Trainee doctors—Senior House Officers | 4 | 3 | 7 |
| Trainee doctors—Registrars | 2 | 3 | 5 |
| Total | 12 | 12 | **24** |

GNCH, Great North Children's Hospital, Newcastle; SMH, St Mary's Hospital, London.

**Box 1    Summary of themes**

**Factors encouraging the use of point-of-care tests**
*Advantages*
► Ease of use and rapid availability of test results (leading to faster decision-making and enhanced patient flow).
► Improved clinical care (resulting from targeted antibiotic treatment, facilitated patient cohorting, closer monitoring and improved communication with caregivers).
► Minimal test invasiveness.
*Other factors*
► Being less experienced.
► Being a paediatric infectious disease specialist > general paediatrician > GP.
► Guidelines recommending their use.

**Factors discouraging the use of point-of-care tests**
► Concerns over test accuracy.
► Failure to resolve diagnostic uncertainty and alter clinical management.
► Preferential reliance on clinical skills and acumen.

## Patient and public involvement

Patients and members of the public were not involved in the design of this study.

## RESULTS

A total of 24 out of 27 invited healthcare workers were interviewed (table 1); two nurses and one trainee doctor did not reply to the invitation. Interviews lasted for 60–75 min. Most participants fully engaged with the interviews and readily shared their views on the topics of interest with little direction from the interviewers. Participants' experience of using POCTs varied at the two hospitals. Although all participants reported routine use of urinary dipsticks, RSV rapid tests and blood gas tests, some participants from St Mary's Hospital recalled also having access to CRP and rapid strep tests in past studies (please see online supplemental file 6 and 7 for description of POCT availability and oversight at the two hospitals).

A range of reasons was given for and against using POCTs. These have been grouped into eight themes (box 1): five pertaining to factors encouraging the use of POCTs (box 2) and three for factors discouraging the use of these tests (box 3) in the management of febrile children.

## Factors encouraging the use of rapid POCTs

Most participants were keen on using POCTs due to their inherent advantages (box 2). One of the main cited advantages was their ease of use and speed with which they can provide results when compared with hospital laboratory-based diagnostics. This was seen to contribute to accelerating decision-making and improving patient flow in busy wards. However, a few participants questioned the magnitude of this impact and noted that these advantages were largely context dependent. In circumstances where laboratories were able to return results within one to two

hours, some felt that the potential benefit in terms of the rapidity of POCTs result was outweighed by the perceived greater accuracy of laboratory-based tests, the possibility to monitor the patient's clinical evolution and the flexibility to see other patients in parallel while awaiting laboratory results.

Urine dipsticks and CRP POCTs were reported to support quicker decision-making on whether or not to prescribe antibiotics or to perform additional diagnostic tests. Rapid RSV testing was valued for assisting with the timely cohorting of children with respiratory infections, thereby limiting the potential for nosocomial transmission. The ability to use rapid POCTs without the help of other doctors or nurses, and to get results independently from the laboratory, was important for many participants, because it allowed for easier repeat testing for monitoring purposes and provided participants with a greater sense of direct control over the care of their patients. Blood gas POCTs were a notable example of this. Additionally, obtaining results rapidly allowed clinicians to quickly communicate results to parents which helped them to justify clinical decisions and to reduce overall parental anxieties, which was felt to have contributed to improved caregivers' experience. Finally, a noted benefit of blood-based POCTs was that they could be performed on finger-prick blood rather than requiring venepuncture, which was seen as more invasive and difficult to perform. This contributed to their acceptability not only to clinicians but also to children and their parents. Aside from the described advantages inherent to POCTs, there were also other factors that influenced the use of POCTs. For doctors, their level of experience appeared to influence their use of diagnostics. There was a prevailing perception that less experienced clinicians used more POCTs as the tests served as learning tools as well as a 'safety net' if their clinical acumen proved inadequate. Moreover, participants also alluded that diagnostics use, including

**Box 2  Direct quotes about the factors that encourage the use of rapid point-of-care tests**

**Advantages**
**Ease of use and rapid availability of test results**
'If it's something the nurses could do really easily and quickly then it would help again, the flow and performance through the department. So let's say it takes 20 minutes to wee [make urinate] a child and it takes a specialist, a nurse, and the doctor…the nurse holding and etc, and you've got a really busy queue, and if you've got a really easy finger prick test that a nurse can do. (ED Paeds Consultant 1)

'Because it [POCTs in general] would just make things a lot quicker. Decision making a lot quicker. Being seen by doctors a lot quicker.' (Nurse 3)

'So, you still might sit on the fence and want to admit and observe but it helps you just maybe make that decision a little bit quicker.' (ED Paeds Consultant 1)

'So speed is probably the main advantage really and by virtue of being less people involved, it's a bit more accessible. You can also do them yourself, so you don't necessarily have to wait for other people to do the test.' (Trainee 8)

**Improved clinical management of febrile children**
**a) More targeted antibiotic treatment**
'So that [urine dipstick] might change your choice of…not only whether you would give them antibiotics, but also what antibiotics you would give.' (Trainee 9)

'I think that [POCTs in general] would help us with our overprescribing but also help reassure us that we are finding the ones that do need antibiotics and getting them to them [the children].' (ED Paeds Consultant 2)

'…I think bloods is probably one of the bigger ones…it just gives a much better indication as to whether or not we would treat with antibiotics or what…like…what kind of route you would go down…' (Nurse 3)

**b) Deciding cohorting and disposition of patient**
'The RSV test, we primarily…, are used for children who are being admitted. So we don't use it as a diagnostic test per se for children in the ED. But they use it for cohorting children if they are admitted to the ward.' (ED Paeds Consultant 5)

'You know that kid where you are a little bit unsure, then the CRP come back at 5. And you were kinda thinking that it was probably ok, then it would influence a bit. Whereas the CRP came back at 60 or 100, then you be like, right they definitely need to come in (be hospitalised) and that sort of stuff.' (ED Paeds Consultant 1)

**Box 2  Continued**

'With the urine [dipstick], I think we're determining whether they need other tests really and whether we need to screen further or look for other sources of infection.' (ED Paeds Consultant 2)

**c) Closer monitoring for clinical changes**
'It's the repeat tests that I think that's the bigger advantage that they allow more easy repeat so you can assess the response to your treatment (…)I think that is the single biggest advantage 'cause it gives you the quantification of the effect of your treatment.' (ED Paeds Consultant 4)

'CRP (…), I think it is useful for monitoring if you've got a patient that is not following the usual course. Like if you got a drop in CRP and a raise again. I think it's very useful' (ED Paeds Consultant 2)

**d) Improved communication with caregivers**
'Parents find it quite stressful how long something like bloods can take. They often come in and ask like how long it's going to be until my bloods are back. At least with urine and blood gas, you can kind of tell them fairly instantly. Like put their mind at rest and things.' (Nurse 1)

'I think parents put great faith in technology. If you have a CRP that is negative, then it's negative. And if it's positive, then it's positive. So I think it's much more black and white way. And so, I think…it can be very convincing for them. So I think it can be helpful in that way.' (Trainee 9)

'I think if somebody [parents] wanted antibiotics and I was sure they didn't [need] and the [rapid] CRP was there, I would be tempted, even though I thought know that it's not necessarily evidence based to do a POCT, and say your CRP is only 6, you don't need antibiotics.' (ED Paeds Consultant 5)

**Minimal test invasiveness**
'I might use that 'cause it's easy to do whereas I wouldn't want to do a proper venous blood test on a child because it's a painful procedure. A child would tolerate a little finger-prick, not like it, but they will tolerate it. Whereas a proper blood test involves lots of screaming, time, big needle, waiting around for hours.' (Gen Paeds Consultant 2)

'Just thinking about, managing a sweaty toddler whose heart can be tachycardic who I think is probably going to be ok, and I'm struggling to get bloods, but if I could do a finger prick and I can get results in ten minutes, if it was a CRP of 100, I could take my time that we've got access and reassure the parents of why that was, whereas if it was 10, I'd say well alright, we don't have to do bloods, we're just going to keep you here for a few hours and see how you are…so I think that would be helpful.' (ED Paeds Consultant1)

**Other factors**
**Training, experience and specialty**

## Box 2  Continued

'When you are a junior doctor you do a bit of more investigations and then you build a bit more of a profile and you get a feel for whether children are more unwell or not. (As a senior,) you probably tend to use less investigations.' (ED Paeds Consultant 1)

'We would have a tendency in the A&E to do more investigations than a GP would…it's all about what you are used…what population that you see. So obviously, if you are an ID [infectious diseases] consultant, then you're seeing all the weird and wonderful so all the other stuff is filtered out by parents and GP and then by A&E and so the population that they see is different.' (ED Paeds Consultant 1)

'If I had a point of care available for other things, I would use it. I would use a CRP, an FBC, I mean I would use everything if I can get it instantly from a finger-prick size sample. I know there is a feeling that probably when you get more senior, that you would sometimes you don't want the information from a blood test but at my level, I would…' (Trainee 5)

### Guidelines
'And so, I think we are very influenced by guidelines because we feel securities with guidelines. And so, to do a POCT, I would do it [if it is in the guidelines] because if I don't do it and it goes wrong, how will I defend myself. And essentially in fever, I already feel vulnerable.' (Trainee 9)

'So I think with a POCT around fever, then you have to have a guideline about how you are gonna use the results. Because especially things like CRP, everyone is going to interpret CRP differently. (…) And so it would have to be a part of a guideline on how to interpret the results.' (Trainee 9)

'If the guideline says to do…if there was a well-respected guideline in place for a particularly clinical scenario that says you should use this test, then of course, I would expect we would have to have a very good reason of why we aren't doing that test.' (ED Paeds Consultant 4)

## Box 3  Direct quotes about the factors discouraging the use of rapid point-of-care test

### Concerns over test accuracy
'It may still be a viral infection and you have a borderline CRP and it may mean you have to go on and do further tests.' (ED Paeds Consultant 2)

'There is a perception that they [POCTs] are less reliable, so often, there's a follow up with the formal lab equivalent…if you are shown a strong evidence base for accuracy of each POCT test, that would be critical for adoption.' (Trainee 11)

'… if there was one test that could tell me if it wasn't bacterial infection, then yup. Fantastic. Something better than CRP. Yeah, it'd definitely be useful for that. If there was a viral diagnostic test that could tell me every single virus. Absolutely. Yeah. Definitely. I think if there were better, more evidence-based tests than yes I think we will definitely use them more.' (ED Paeds Consultant 5)

'I guess you'll have to do some research into that in terms of how to validate that…because sometimes I feel like a lot of the research…are actually done on…adults rather than children.' (Trainee 7)

### Failure to resolve diagnostic uncertainty and alter clinical management
'I don't think any of the POCTs necessarily give you a diagnosis unless you've got a urine dipstick that is positive for urine infection. Umm, so I don't think they really change your diagnosis as such.' (Trainee 2)

'So I think that if I'm at the point where I want more information and I want to know what the CRP, I automatically want to observe them for a few hours, so it doesn't actually make that much of a difference to my working practice because I put less emphasis on the test than I do on the observations.' (Trainee 9)

'I want something that's a binary. Because I think what you might find with a lot of tests, it is a lot of grey. And it will give you an answer that, it could be a virus, it could be a bacteria; we're not really that sure. In which case it's useless as another blood test. Quicker, but otherwise no better. Because that's what our CRP, our white cells show, and we are still left in the situation.' (Gen Paeds Consultant 2)

'I would be far more in favour of a test that could predict the likelihood of significant deterioration rather than one that could tell me whether something is viral or bacterial.' (ED Paeds Consultant 4)

'So, if you take an NPA [nasopharyngeal aspirate], which is how we test for viral respiratory panels here, you're testing for 10 viruses. If you only take the RSV rapid test, you are only testing for RSV. And you still need to get a sample from the child. If you are getting a sample then why not just get the whole thing?' (Trainee 5)

POCTs, varied according to their perceptions of the risk for severe infections, which itself was dependent on the clinician's specialty training and the population seeking care. Some participants believed that GPs used less diagnostics than emergency department paediatricians, who in turn use less diagnostics than paediatric infectious disease specialists. This was thought to be because the patient population seen by the three types of clinicians differed, with GPs seeing proportionally fewer children with severe infections.

Finally, clinical guidelines were also seen as important. Participants reported that guidelines including recommendations on the use of the tests and the interpretation of results in children would be highly influential in their decision to use POCTs. This was because guidelines were perceived as both providing reassurance of best practices and protection against malpractice.

## Box 3   Continued

'I guess the thing that I would want to know is that: are they [POCTs] likely to change my management here and how?...in a way that is going to be clinically significant. So, for example, a throat swab, if I'm going to prescribe antibiotics and the throat swab confirmed that then that is great, but that hasn't really changed anything.' (ED Paeds Consultant 4)

### Preferential reliance on clinical acumen
'Just the basic clinical skills that doctors...I think POCTs supplement what we do as doctors rather than replace it. I don't think they can be used in isolation.' (ED Paeds Consultant 2)

'If I wasn't worried, I would trust my judgement because sometimes if you do further test to reassure a parent and it's not needed, you are putting a child through invasive tests, but you are potentially going to find something that might not be helpful in the bigger picture...Where I can, I like to manage on my clinical judgement in looking at the child and my observational skills.' (ED Paeds Consultant 2)

'Like I said, I think basically a lot of what I do is based on history and examination...the essential skills that doctors need to know. I trust my history and examination quite a bit.' (Trainee 10)

### Factors discouraging the use of rapid POCTs
Several participants were concerned about the accuracy of the test results (box 3). It was commonly perceived that there were discrepancies in relation to the patient's clinical presentation and/or when compared with laboratory tests. These concerns undermined clinicians' trust in POCTs, and often resulted in subsequent confirmatory laboratory tests to be performed prior to making a clinical decision. Many also cited the lack of strong evidence supporting the accuracy of some POCTs for paediatric use, some noting that existing POCTs such as CRP had predominantly only been validated in adults.

Another notable reason for hesitancy about some POCTs was their perceived inability to adequately address diagnostic uncertainty. Some participants mentioned that with current POCTs that provide numerical results (e.g. CRP levels) instead of binary results ('positive/negative'), the tests were unable to provide a definitive clinical answer when results were within an 'intermediate' range. Other participants felt that current POCTs were not able to reliably predict which initially well-appearing febrile children were at risk of clinical deterioration and therefore required hospitalisation. This was perceived to be an important feature of diagnostics for managing fever in children. It was also commonly noted that some POCTs were only designed to detect a single pathogen such as RSV, which was of limited utility compared with the range of different pathogens that were detectable from laboratory-based panels. Altogether, this failure to resolve diagnostic uncertainty was seen to limit the potential for

POCTs to alter clinical decisions. In addition, even when POCTs were perceived as accurate, it was felt that their results might not prove consequential enough to warrant a change in the patient's overall clinical management. An example of this was the rapid Group A strep test. While the test was acknowledged to be sensitive and specific, some participants felt that they would make a decision to prescribe antibiotics irrespective of whether a test confirmed the presence of *Group A streptococcus* or not. However, the results of other POCTs, such as urine dipsticks, greatly influenced clinical management for many participants.

Overall, in view of the perceived limitations of POCTs, participants expressed a preference for prioritising their clinical skills (history, examination, observation of patients) and acumen over the reliance on diagnostics in guiding their clinical management of febrile children.

### Comparison of views between types of participants
The views expressed by different types of participants were broadly similar. However, there was a trend towards some differences emerging, mainly between trainees versus consultants and nurses. Specifically, consultants stressed the importance of relying on their clinical acumen, more so than trainees. They also felt that trainees used more POCTs, a sentiment was also shared by nurses, but this was seldom reported by trainees. Trainees more often mentioned how CRP results can appear normal for a child at the beginning of a serious infection, while consultants seemed more concerned with the difficulty of interpreting intermediate CRP results. The potential contribution of POCT usage in improving antibiotic stewardship was mentioned widely but generally with stronger emphasis by consultants and nurses than trainees. Finally, nurses mentioned their willingness to perform more POCTs independently of doctors in the future. While consultants and trainees rarely commented on who should be performing the tests, when they did, they were not against nurses using POCTs independently.

### DISCUSSION
This study provides insight into what hospital-based doctors and nurses in England think of POCTs in relation to managing children with suspected infection, a topic on which little has previously been published. Within the two hospitals, we interviewed a diverse range of healthcare providers who have responsibility for deciding on the use of POCTs including both consultants in general paediatrics and emergency paediatrics, paediatric trainees with different levels of experience and nurses. A range of reasons for and against the use of POCTs was identified.

The advantages of POCTs over laboratory-based tests in terms of their ease of use and the rapid availability of results were widely recognised. Other reported potential benefits included faster clinical decision-making; improved monitoring of patients; more targeted use of antibiotics; enhanced patient flow; the rapid cohorting

of inpatients with respiratory infection and improved communication with parents. However, these advantages were less evident when laboratories were able to return results within a few hours. Being able to perform blood-based POCTs on finger-prick samples was also seen as less invasive and, therefore, more acceptable to clinicians, parents and children compared with tests requiring venepuncture. Guidelines were also perceived to be important in guiding patient management, including the use of POCTs.

However, participants also mentioned a number of important limitations of POCTs. These include their perceived inaccuracy, their inability to predict risks of clinical deterioration and the limited scope of diagnostic information from certain tests. Some POCTs were not seen as being able to solve diagnostic uncertainty and others were perceived as having a limited role in guiding or changing overall clinical management. Altogether, these perceived shortcomings reinforced the overarching view that clinicians should trust their clinical acumen more than diagnostics.

The only other study that we were aware of which specifically explored the views of paediatricians and nurses about rapid POCT usage in children, found similar results. Participants valued their ease and speed of use, which enabled early treatment, cohorting, decreasing antibiotic prescription and better communication with parents.[20] Some participants were also concerned about the accuracy of the tests and about the limited changes in clinical management enabled by POCTs.

Some studies exploring the views of GPs reported similar advantages and concerns, especially regarding the accuracy of tests.[12 14 20 24–26] However, there were conflicting results in terms of the perceived invasiveness of POCTs: two studies found that GPs would not use blood-based POCTs because they felt finger pricking was too invasive in children,[12 25] while another study reported that GPs perceived the pain to be short-lived and recounted that children were actually interested by the use of rapid tests during consultations.[27] Other GPs felt it was easy to persuade parents that antibiotics were not needed compared with adult patients, limiting the need to use POCTs for this purpose.[25] Some GPs reportedly also feared that the use of POCTs could actually increase antibiotic prescription.[25] Finally, in contrast to our finding that for some paediatricians, the perceived limitations of POCTs reinforced their preference of relying on their clinical acumen over diagnostics, some of the studies with GPs reported that there were concerns that the perceived advantages of POCTs could lead to the tests being over adopted and used, which could potentially undermine clinical acumen in the long term.[13 14 25]

Of note, national and local paediatric guidelines were seen to be important in guiding patient management, including the use of diagnostics. At the time of the study, the use of urine dipstick and blood gas analysis were mentioned in relevant guidelines for managing children with suspected infections. These guidelines were widely available in hospital acute care settings in England. However, to date, neither CRP POCT nor rapid strep tests have been recommended for the management of acute childhood infections by the National Institute for Health and Care Excellence and neither were available in the two hospitals at the time of this study.

The study had some limitations and also highlighted a number of important areas for further research. First, as the study aimed to explore the role of POCTs used in febrile children rather than focusing on one or two specific tests, it was sometimes not possible to decipher whether the participants' comments were referring to specific POCTs or POCTs in general. Given the substantial differences in the attributes and roles of different POCTs, deeper exploration about specific POCTs would be helpful.

Second, several members of our research team are active paediatricians with varying experience of POCTs. However, we have trained and worked in a wide variety of countries and settings and hold a wide range of views about POCTs ourselves. Moreover, two public health specialists with limited experience in diagnostics or paediatrics research conducted the interviews, contributed to the data analysis, interpretation and drafting of results. Therefore, although as in any qualitative research, there is an inherent risk of subjectivity in the interpretation and presentation of the results, we believe that the breadth of views and backgrounds in the team have helped to minimise this risk.

Third, the scope of this study was restricted to two teaching hospitals in England and did not capture the views of paediatricians in district general hospitals, nor GPs. Further qualitative research involving other key stakeholders, such as clinicians in other subspecialties and those based in district general hospitals or in the community, along with parents and children themselves, would be very informative. As the study was limited to England, we could not make any comparisons with other countries where the culture of clinical decision-making and availability of diagnostic testing may differ. As the use of POCTs seems to vary greatly across countries, it would be compelling to carry out similar studies elsewhere. It would also be useful to elucidate in which patients, as well as where within the care pathway, specific POCTs are likely to have the most impact. Finally, this study was carried out in the pre-COVID-19 era. Since then, the diagnostics landscape has changed dramatically; it would informative to carry out a similar study now, as the views about POCTs may have evolved since.

## CONCLUSION

In this study conducted in two university teaching hospitals in England, participants expressed mixed opinions about the utility of current POCTs in the management of febrile children. While the many potential advantages of POCTs were well recognised, concerns about their accuracy and their failure to solve diagnostic uncertainty and alter clinical management limited the perceived utility of current POCTs in this setting. Understanding the clinical decision-making process and the concerns and needs

of clinicians in different settings will be critical in the development of new tests and in evaluating the potential impact on clinical management and other outcomes. Further studies in different countries and healthcare settings in the post-COVID-19 era, should help to inform this understanding.

**Author affiliations**
[1]Clinical Research Department, Faculty of Infectious and Tropical Disease, London School of Hygiene and Tropical Medicine, London, UK
[2]Patient Safety Translational Research Centre, Institute of Global Health Innovation, Department of Surgery & Cancer, Imperial College, London, UK
[3]Department of Paediatric Immunology, Infectious Diseases & Allergy, Great North Children's Hospital, Newcastle upon Tyne, UK
[4]Paediatric Emergency Department, St Mary's Hospital Imperial College Healthcare NHS Trust, London, UK
[5]Section of Paediatric Infectious Diseases, Faculty of Medicine, Imperial College London, London, UK
[6]Department of Paediatric Infectious Disease, St Mary's Hospital Imperial College Healthcare NHS Trust, London, UK

**Acknowledgements** The authors wish to thank all the participants who enthusiastically contributed their valuable time and thoughts to this research project. We are also thankful to Dr Emma Lim for her support.

**Collaborators** Dr Emma Lim, Great North Children's Hospital, Newcastle upon Tyne

**Contributors** JED, EL and QL contributed equally to this paper. JED and SY conceived of the study. SY had overall responsibility. All authors input into the design of the study and the study materials. ME, IM, RN and SY identified participants. EL and QL conducted and transcribed the interviews. EL and QL analysed the data and drafted the manuscript under JED's supervision. EL, JED, and SY made substantial revisions. All authors provided input into drafts of the manuscript and agree on the contents of the final version.

**Funding** JED, ME and SY are supported by PERFORM, a consortium funded by the European Union's Horizon 2020 programme, under grant agreement number 668 303. RGN was supported by NIHR Academic clinical fellowship (CF- 2015-21-016) and lectureship (CL-2018-21-007) award programme. The funding bodies did not take part in the design of the study and data collection and did not take part in the data analysis and interpretation of results.

**Competing interests** None declared.

**Patient consent for publication** Not required.

**Provenance and peer review** Not commissioned; externally peer reviewed.

**Data availability statement** Data are available upon reasonable request. The datasets generated and/or analysed during the current study will be made available upon reasonable request through the London School of Hygiene and Tropical Medicine data depository (LSHTM Data COMPASS).

**ORCID iDs**
Edmond Li http://orcid.org/0000-0001-8209-4490
Juan Emmanuel Dewez http://orcid.org/0000-0002-5677-8968
Shunmay Yeung http://orcid.org/0000-0002-0997-0850

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
