## [Reviewer comments · BMJ Open]

ARTICLE DETAILS

TITLE (PROVISIONAL)	The role of Point-of-Care-Tests in the management of febrile children: A qualitative study of hospital-based doctors and nurses in England
AUTHORS	Dewez, Juan; Li, Edmond; Luu, Queena; Emonts, Marieke; Maconochie, Ian; Nijman, R; Yeung, Shunmay

VERSION 1 – REVIEW

REVIEWER	Muthukumar Sakthivel Sidra Medicine
REVIEW RETURNED	18-Nov-2020

GENERAL COMMENTS	I had reviewed the article previously, thanks for amendments.
---

REVIEWER	Fiona Wood Cardiff University, Division of Population Medicine
REVIEW RETURNED	30-Nov-2020

GENERAL COMMENTS	This is an interesting paper, but there are a number of concerns I have before I would recommend it for publication. Some of these are relatively minor. Page 4, line 25, Antibiotic prescribing (not antibiotics prescribing) line 26 Optimise the use of medical resources? line 40 - what do you mean by cohorting? This is also mentioned on page 14 line 251 line 46 - not sure you should be saying POCTs' inability.... doesn't sound quite right page 8, line 127, it's not clear how the participants were selected (or approached). Did the researchers have a list of potential respondents. Were the participants invited by email? letter? in person? Page 9, the authors say their sample size was based on other qualitative studies of healthcare workers perceptions, but this is very broad. The references you give are relevant, but I think the text needs tweaking. The authors mention data saturation, but you give no justification for reaching the point of data saturation nor how you assessed you had reached this point. It's not clear how long the interviews were (on average) It is a bit disappointing that there was no PPI involvement You have University ethics approval. I realise that no NHS ethics approval required as no patients were involved in study, but what
---

	about research governance for the 2 hospital Trusts? Can you confirm you have all necessary approvals to conduct the research with NHS staff in NHS settings? There is also no mention of appropriate written informed consent. It is not currently clear where the interviews took place. (eg. did they take place on NHS site at participants' usual place of work? 24 out of 27 invited participants is a good response rate, but I am a bit worried about this, given the lack of information about how participants were selected and approached. There are times within the results section that the written language is not so good. So for example, page 11 lines 177-179 need amending for grammar. Similarly, the sentence "Aside from advantages inherent to POCTs, other factors influenced the use of POCTs." needs rephrasing. Page 15, line 290 POCTs were unlikely... - should this be are unlikely? The document also needs proof reading. There are some extra full stops etc, inappropriate use of tenses etc. The paper is generally of a good written standard, but there are some poor sentences. Overall I wonder if the authors have tried to cover too much with discussing a number of different types of POCTS, some of which not at all invasive, others are. Of course these POCTs have differing levels of accuracy. If accuracy is one of the main concerns, then surely it's necessary to tease this out further and lumping all POCTS together is a problem here. The paper would benefit from some discussion of reflexivity. How do the researchers' own beliefs and ideas influence the research question, the analysis and the presentation of findings? It is not enough just to mention this in the section on conflicts of interest. There are things the authors flag up in the abstract (for example how POCTs help with parent communication) that are not discussed in the results. Raising things in the abstract, with no evidence in the main paper is not appropriate.
--	---

VERSION 1 – AUTHOR RESPONSE

Response to reviewer 1

Comments	Response
I had reviewed the article previously, thanks for amendments.	Thank you very much for re-assessing our manuscript.
Additional comment from reviewer 1 attached to the email from the editorial office: Despite the limitations such as sample size and diversity, this study provides with valuable information on the perceptions of health care professionals on the role of POCTs in the assessment of febrile children.	Thank you, we hope we have adequately acknowledged the study limitations.

Response to reviewer 2

Thank you again for your assessment. We are grateful for the opportunity to add detail that we had not presented earlier in the interest of limiting the word count. We believe that the additional information significantly improves the quality and transparency of our reporting.

Comments	Response
Page 4, line 25, Antibiotic prescribing (not antibiotics prescribing)	Thank you. We have corrected this (line 28).
line 26 Optimise the use of medical resources?	Thank you. We agree that this is not clear – we have deleted. (line 27- 28).
line 40 - what do you mean by cohorting? This is also mentioned on page 14 line 251	By “cohorting” we mean where hospitalised patients with the same infections are placed in the same dedicated ward in order to reduce the spread to others in the hospital eg paediatric RSV infection. We have described this in more detail (line 42)
line 46 - not sure you should be saying POCTs' inability.... doesn't sound quite right	‘Inability’ has been removed. We have revised (lines 46-48): “However, in addition to their perceived inaccuracy, participants were concerned about POCTs not resolving diagnostic uncertainty or altering clinical management.”
page 8, line 127, it's not clear how the participants were selected (or approached). Did the researchers have a list of potential respondents. Were the participants invited by email? letter? in person?	We agree that this was not described in much detail. We have provided more details to clarify our approach (lines 128-133). “A sampling matrix was developed specifying the targeted number of participants of each healthcare worker category (consultant paediatrician, paediatric trainee with three or more years training, paediatric trainees with less than three years training, and ED nurses). As members of staff of the two hospitals, SY, RN, IM, ME contacted available paediatric staff who fit the inclusion criteria to ascertain their interest in being interviewed for the study. Those who agreed, were then followed-up by email by the interviewers ...”
Page 9, the authors say their sample size was based on other qualitative studies of healthcare workers perceptions, but this is very broad. The references you give are relevant, but I think the	We have replaced “Comparable” with “a sample size which is in line with other qualitative studies” (line 135-136).

text needs tweaking.	
The authors mention data saturation, but you give no justification for reaching the point of data saturation nor how you assessed you had reached this point.	We agree that this is important and have provided further details: “Identifying the actual point of data saturation was important to make best use of the available time and resources and to minimise the disruption caused by diverting healthcare providers from their normal clinical activities. During the regular debriefings, EL, QL, and JED identified whether any new ideas had been raised by the participants. Once no new ideas had emerged from the last several interviews, the authors agreed that data saturation has been reached.” (lines 137-141)
It's not clear how long the interviews were (on average)	The interviews lasted 60-75 minutes. We have added this to the manuscript: (line 163).
It is a bit disappointing that there was no PPI involvement	We understand the disappointment, however, the aim of the study was specifically to explore the views of healthcare providers. We believe that seeking the views of children and parents is very important and indeed our colleagues in Newcastle have ongoing work with both young people and parents. We were concerned that trying to include the views of both health care providers and parents in a single study would have resulted in a loss of depth and focus.
You have University ethics approval. I realise that no NHS ethics approval required as no patients were involved in study, but what about research governance for the 2 hospital Trusts? Can you confirm you have all necessary approvals to conduct the research with NHS staff in NHS settings?	We have full ethical and administrative approval for this study. Ethical approval:  • The NHS has confirmed that ethical approval was not required as no patients were involved in this study. • We obtained ethical approval from the London School of Hygiene and Tropical Medicine (ref: 15040-15088) We obtained Health Research Authority approval (Ref: IRAS 248723) Administrative approval: We obtained administrative approvals from:  • The Joint Research Compliance Office of Imperial College Healthcare NHS Trust (ref: 18SM4662) • The Research & Development office of the Newcastle Upon Tyne Hospitals NHS Foundation (ref: 248723)

	We added this information to the manuscript (lines 155-158).
There is also no mention of appropriate written informed consent.	Apologies for this important omission. We confirm that all participants provided written informed consent after reading the participant information sheet. We have added this information to the manuscript (line 133-134).
It is not currently clear where the interviews took place. (eg. did they take place on NHS site at participants' usual place of work?	The interviews all took place in the clinical workplace of the two hospitals during normal work hours. This has been added to the manuscript (line142-143).
24 out of 27 invited participants is a good response rate, but I am a bit worried about this, given the lack of information about how participants were selected and approached.	We have provided additional information on how participants were selected and approached (lines 122-134).
There are times within the results section that the written language is not so good. So for example, page 11 lines 177-179 need amending for grammar. Similarly, the sentence "Aside from advantages inherent to POCTs, other factors influenced the use of POCTs." needs rephrasing. Page 15, line 290 POCTs were unlikely... - should this be are unlikely?	Apologies for these errors and thank you for identifying them. We have amended it with the following: "Finally, a noted benefit of blood-based POCTs was that they could be performed on finger-prick blood rather than requiring venepuncture which was seen as more invasive and difficult to perform." (lines 192-195). "Aside from the described advantages inherent to POCTs, there were also other factors which influenced the use of POCTs" (lines 196-197). The last sentence has been deleted during our editing.
The document also needs proof reading. There are some extra full stops etc, inappropriate use of tenses etc. The paper is generally of a good written standard, but there are some poor sentences.	We have carefully proofread the manuscript and tried to identify and correct all the typological and grammatical errors. We have also re-written a number of sections to improve the clarity.
Overall I wonder if the authors	This was an important matter of debate in the research team

have tried to cover too much with discussing a number of different types of POCTS, some of which not at all invasive, others are. Of course these POCTS have differing levels of accuracy. If accuracy is one of the main concerns, then surely it's necessary to tease this out further and lumping all POCTS together is a problem here.	during the conception of the study. We had to find a balance between focussing more in depths on a single/few POCTS vs. being more inclusive of more POCTS which could be used in febrile kids. We decided that it would be more informative for clinicians and policy makers to have a wider scope of POCTS to be examined, and we decided to use the latter approach. This is recognised as a limitation in Strengths and limitations section (lines 64-65), and in the discussion section (lines 297-299).
The paper would benefit from some discussion of reflexivity. How do the researchers' own beliefs and ideas influence the research question, the analysis and the presentation of findings? It is not enough just to mention this in the section on conflicts of interest.	We agree with this suggestion and have made additions to the discussion section of the manuscript to reflect this (lines 301-307). “Secondly, several members of our research team are active paediatricians with varying experience of rapid POCTS. However, we have trained and worked in wide variety of countries and settings, and hold a wide range of views about rapid POCTS ourselves. Moreover, two public health specialists with limited experience in diagnostics or paediatrics research conducted the interviews, contributed to the data analysis, interpretation, and drafting of results. Therefore, although as in any qualitative research, there is an inherent risk of subjectivity in the interpretation and presentation of the results, we believe that the breadth of views and backgrounds in the team have helped to minimise this risk.”
There are things the authors flag up in the abstract (for example how POCTS help with parent communication) that are not discussed in the results. Raising things in the abstract, with no evidence in the main paper is not appropriate.	We fully agree and apologise for this. We have re-checked and made amendments to ensure that all these sections are consistent.

VERSION 2 – REVIEW

REVIEWER	Fiona Wood Cardiff University, Division of Population Medicine
REVIEW RETURNED	01-Mar-2021

GENERAL COMMENTS	Thank you for making amendments to the paper. Clearly quite a lot of changes have been made and I agree the paper is now better. I can't see if a COREQ (reporting criteria for qualitative research) has been completed and the editorial office may require this.
---

VERSION 2 – AUTHOR RESPONSE

We had originally uploaded the SRQR checklist as a supplement. However as per request, we have now also filled in and uploaded the COREQ check list.